# Surf Redfish-Based ZnO-NPs and Their Biological Activity with Reference to Their Non-Target Toxicity

**DOI:** 10.3390/md21080437

**Published:** 2023-08-02

**Authors:** Ahmed I. Hasaballah, Hussein A. El-Naggar, Ibrahim E. Abd-El Rahman, Fatimah Al-Otibi, Reham M. Alahmadi, Othman F. Abdelzaher, Mohamed H. Kalaba, Basma H. Amin, Mohamed M. Mabrouk, Ahmed G. A. Gewida, Marwa F. Abd El-Kader, Mostafa A. Elbahnasawy

**Affiliations:** 1Zoology and Entomology Department, Faculty of Science, Al-Azhar University, Nasr City 11884, Egypt; hu_gar2000@azhar.edu.eg (H.A.E.-N.); o.fadl@yahoo.com (O.F.A.); 2Department of Plant Protection, Faculty of Agriculture, Al-Azhar University, Cairo 32897, Egypt; dribrahimelsayed2014@azhar.edu.eg; 3Department of Botany and Microbiology, College of Science, King Saud University, Riyadh 11451, Saudi Arabia; falotibi@ksu.edu.sa (F.A.-O.); realahmadi@ksu.edu.sa (R.M.A.); 4Botany and Microbiology Department, Faculty of Science, Al-Azhar University, Nasr City 11884, Egypt; dr.m_kalaba@azhar.edu.eg (M.H.K.); mostafa.elbahnasawy@azhar.edu.eg (M.A.E.); 5Regional Center for Mycology and Biotechnology, Al-Azhar University, Cairo 11884, Egypt; basmafarag.18@azhar.edu.eg; 6Fish Production Department, Faculty of Agriculture, Al-Azhar University, Cairo 32897, Egypt; mabrouk3m@azhar.edu.eg (M.M.M.); a.gewida@azhar.edu.eg (A.G.A.G.); 7Central Laboratory for Aquaculture Research, Sakha Aquaculture Research Unit, Department of Fish Diseases and Management, A.R.C., Kafrelsheikh 33516, Egypt

**Keywords:** sea cucumber, surf redfish, ZnO-NPs, antibacterial, larvicidal, adulticidal, non-target organism

## Abstract

The marine environment is a rich source of bioactive compounds. Therefore, the sea cucumber was isolated from the Red Sea at the Al-Ain Al-Sokhna coast and it was identified as surf redfish (*Actinopyga mauritiana*). The aqueous extract of the surf redfish was utilized as an ecofriendly, novel and sustainable approach to fabricate zinc oxide nanoparticles (ZnO-NPs). The biosynthesized ZnO-NPs were physico-chemically characterized and evaluated for their possible antibacterial and insecticidal activities. Additionally, their safety in the non-target organism model (Nile tilapia fish) was also investigated. ZnO-NPs were spherical with an average size of 24.69 ± 11.61 nm and had a peak at 350 nm as shown by TEM and UV-Vis, respectively. XRD analysis indicated a crystalline phase of ZnO-NPs with an average size of 21.7 nm. The FTIR pattern showed biological residues from the surf redfish extract, highlighting their potential role in the biosynthesis process. DLS indicated a negative zeta potential (−19.2 mV) of the ZnO-NPs which is a good preliminary indicator for their stability. ZnO-NPs showed larvicidal activity against mosquito *Culex pipiens* (LC_50_ = 15.412 ppm and LC_90_ = 52.745 ppm) and a potent adulticidal effect to the housefly *Musca domestica* (LD_50_ = 21.132 ppm and LD_90_ = 84.930 ppm). Tested concentrations of ZnO-NPs showed strong activity against the 3rd larval instar. Topical assays revealed dose-dependent adulticidal activity against *M. domestica* after 24 h of treatment with ZnO-NPs. ZnO-NPs presented a wide antibacterial activity against two fish-pathogen bacteria, *Pseudomonas aeruginosa* and *Aeromonas hydrophila*. Histopathological and hematological investigations of the non-target organism, Nile tilapia fish exposed to 75–600 ppm ZnO-NPs provide dose-dependent impacts. Overall, data highlighted the potential applications of surf redfish-mediated ZnO-NPs as an effective and safe way to control mosquitoes, houseflies and fish pathogenic bacteria.

## 1. Introduction

The development of nanoparticles has been trending in the recent decade due to their benefits not only for industrial applications but also for medicinal and agricultural purposes. A wide range of approaches, including physical, chemical, and biological, have been addressed for synthesis of nanoparticles with high stability and solubility. Among nanoparticle synthesis methods, biological ones have been introduced as a sustainable, ecofriendly and cost-effective method [1]. In the last few years, zinc oxide nanoparticles (ZnO-NPs) have attracted much interest due to the various outstanding characteristics and extended applications they have. ZnO-NPs are versatile metal oxide nanoparticles and are ranked as the third most commonly used metal nanoparticles with numerous electrical, catalytic, and optical properties [2,3,4]. The main advantage of ZnO-NPs is the excellent antimicrobial activity they offer [5]. They are also used with topical medications to accelerate wound healing [6].

The marine environment hosts a huge diversity of organisms that evolved to live in challenging and harsh conditions [7,8], reflecting the enormous by-products and derivatives with unique biological properties that could be beneficial in wide industrial and biotechnological applications [9]. Until now, the marine environment has been considered an unexplored source for promising components of biotechnological interest [5,10]. Recently, growing attention has been focused on discovering natural products and derivatives from natural sources due to their safety, high-nutrition contents of trace elements, and unique bioactive compounds that exhibit beneficial health effects [11,12]. The sea cucumber is a marine invertebrate with important economic value due to its highly valuable nutrients, including vitamins, minerals, and amino acids; the nutrient content of the sea cucumber consists of 86% protein, 3–5% carbohydrates, and 1–2% fat [13]. Lately, investigations of sea cucumbers have attracted more attention due to their promising effectiveness against various diseases, such as cancer, asthma, hypertension, rheumatism, and degenerative diseases [14]. Sea cucumbers contain eicosapentaenoic acid (EPA) and docosahexaenoic acid (DHA), which are considered the most important omega-3 fatty acids which play a role in the development of brain nerves, wound healing, and antithrombotic [15]. Furthermore, sea cucumber also contains bioactive ingredients with antihypertensive [16], antibacterial, antifungal [17] and anti-cancer properties [14].

Mosquitoes are amongst the most widespread insects globally. They are characterized by their facultative host feeding, for example, they feed on both humans and animals, which increases the transmission of disease pathogens and makes them a critical threat to public health [18]. *Culex pipiens* is the most common, widespread and predominant vector in the urban environment particularly in Africa due to the availability of its ideal environmental conditions there. It can be found in different types of water streams, such as polluted sites, drains, septic tanks, and puddles; additionally, they are the main vector of lymphatic filariasis, West Nile Virus, and Rift Valley fever [19]. Muscid flies act as mechanical vectors for many disease pathogens. The housefly *Musca domestica* is a well-known livestock pest of public health importance. The biology of these flies makes them ideal organisms to carry and disseminate human and animal pathogens, such as helminth parasites, protozoan cysts, viruses, and bacteria [20]. Additionally, it has been implicated in the transmission of viruses [21], bacteria, such as *Helicobacter pylori* [22], and many other disease pathogens. Due to the harmful effects of chemically synthetic insecticides and the developed resistance to these chemical substances, an alternative source with promising larvicidal and/or adulticidal properties is highly required. 

In response to the above-mentioned challenges, the present study was conducted to develop an ecofriendly, cost-effective, and non-toxic approach for preparing ZnO-NPs using a sea cucumber, surf redfish, and aqueous extract as a reducing agent. The potential therapeutic properties and non-target toxicity of biosynthesized ZnO-NPs were also investigated.

## 2. Results and Discussion

### 2.1. Identification and Specification of Collected Samples

The sea cucumber *Actinopyga mauritiana* (Quoy & Gaimard, 1834), commonly known as the surf redfish, is a widespread species in the Indo-Pacific and the Red Sea. Surf redfish reach a maximum width of 10 cm and a length of 220 to 350 mm with an elongated body. The body wall is tough and leathery; it can reach about 6 mm thick. The bivium is dark brown or orange in color, has occasional white specks, is sometimes wrinkled, and is broader in the center and tapered toward both ends. The bivium is also covered with slender and long papilles, which are usually brown or dark orange in hue. The trivium is white in color and covered with numerous thick podia. The mouth is ventral, encircled with twenty-five short and strong tentacles, with a huge collar of long papillae at their base. Unlike other sea cucumbers, the pinkish cuvierian tubules are never expelled. Surf redfish are more active during the day, live near outer reef flats and fringe reefs, and can be found in the range of 0–50 m (Figure 1). 

### 2.2. Characterization of Biosynthesized ZnO-NPs 

In the present study, an aqueous extract of surf redfish was used to reduce aqueous Zn^2+^ to ZnO-NPs. A preliminary proof of reduction was detected by the changing of reaction color to a cloudy and milky-color mixture, followed by the development of white to off-white precipitates. Fabrication of ZnO-NPs was initially witnessed as a white haziness, and then deposited in the bottom of the reaction container [5]. Scanning the reaction end-product with UV–Vis spectroscopy (290–710 nm) has shown spectra with a prominent peak at 350 nm, confirming the reduction of Zn^2+^ and formation of ZnO-NPs. Reduction of Zn^2+^ to ZnO-NPs was suggested as a response to the action of biological components and/or functional groups in the surf redfish aqueous extract (Figure 2A). ZnO-NPs usually exhibit a peak between 350 nm and 390 nm in the UV–Vis absorption spectrum as shown by several studies [23,24,25,26]. Morphological examination of ZnO-NPs showed particle sizes averaged at 24.69 ± 11.61 nm as determined with transmission electron microscopy (TEM). TEM micro images of ZnO-NPs showed mostly unaggregated spherical particles with a broad size distribution, a range from 9.0 nm to 50.1 nm, and the predominate size was 22 nm (Figure 2B). 

X-ray diffraction (XRD) analysis of the ZnO-NPs revealed a crystalline phase with a given average size of 21.7 nm as generated with the Debye–Scherrer equation. In addition, the XRD pattern profiled strong peaks at 31.66, 34.21, 36.13, 47.61, 56.46, 62.53, and 67.82, which were categorized as planes (100), (002), (101), (102), (110) (103), and (112) (Figure 2C). These peaks corresponded well with the ZnO criteria from the Joint Committee on Power Diffraction (JCPD), file number (00-005-0664). Therefore, the XRD pattern of ZnO-NPs indicated a fine hexagonal crystalline structure that agrees with this reference model. Furthermore, ZnO-NPs have shown high phase purity free of impurities. Regarding the self-aggregated affinity that most nanomaterials have, stability is a crucial parameter to consider for the newly formed nanoparticles. Nanoparticle stability is directly related to the zeta potential (i.e., surface potential) [27]. In this study, the zeta potential value of ZnO-NPs in a colloidal solution is shown in Figure 2D. ZnO-NPs have shown a negative zeta potential value (–19.2 mV), indicating moderate stability. Fourier transform infrared spectroscopy (FTIR) analysis was carried out to examine the possible involvement of biological molecules from the surf redfish aqueous extracts in the synthesis of ZnO-NPs. FTIR data showed substantial absorption spectra ranging from 400 to 4000 cm^−1^ (Figure 2E). The FTIR spectral analysis displayed absorption peaks at 3420, 2280, 2040, 1620, 1410, 1060, and 580 cm^−1^. The characteristic absorption band at 580 cm^−1^ (ZnO bond) confirmed the formation of ZnO. The sharp absorption peak at 1620 cm^−1^ corresponds to the carbonyl (C=O) stretching vibration of the amide I [28,29]. The band at 1410 cm^−1^ could be attributed to stretching of C–C groups. The two absorption bands at 2280 cm^−1^ are assigned to the C≡C group [30]. The peak at 1060 cm^−1^ could be attributed to stretching C–O, while the broad peak at 3420 cm^−1^ corresponds to the vibration of O–H stretching group in alcohols and phenols [31]. 

### 2.3. Insecticidal Activity

Data obtained in (Table 1) presents a gradually increasing pattern in the mortality percentages of *C. pipiens* larvae induced with ZnO-NPs. Tested concentrations showed strong activity against the 3rd larval instar. The lowest concentration tested (2 ppm) revealed a non-significant effect on larval mortality (*p* = 0.836), while the highest tested concentration (32 ppm) reduced the survival of larvae to about (82.4 ± 1.6%). Probit analysis revealed that the LC_50_ value was (15.412 ppm) and the LC_90_ value was (52.745 ppm). Tested concentrations were significantly different (*d.f.* = 4, *p* < 0.05 and χ^2^ = 11.065). Zinc acetate and ZnO-NPs (without *A. mauritian*) revealed no larval mortality. To the best of our knowledge, there are very few studies on the effects of metal NPs synthesized from marine organisms against mosquito vectors. Of similar lethal concentrations obtained here, Vinotha et al. [32] found that *Elettaria cardamomum* ZnO-NPs were a highly potential agent against *C. tritaeniorhynchus* (LC_50_ = 15.09 μg/mL, LC_90_ = 29.70 μg/mL). Alike, Hasaballah et al. [5] found that ZnO-NPs biosynthesized using the marine sponge *Spongia officinalis* recorded LC_50_ values of 31.82 ppm against *C. pipiens* larvae. Recently, biosynthesised ZnO-NPs using the sea cucumber *Holothuria impatiens* induced much higher larval toxicity against *C. pipiens* larvae [33].

The adulticidal activity of tested ZnO-NPs against *M. domestica* was evaluated using a topical assay. Data obtained in (Table 2) revealed that different doses induced adulticidal activity after 24 h of treatment, particularly when higher doses were applied. Probit analysis revealed that the obtained LD_50_ value was (21.132 μg/adult) and the LD_90_ value was (84.930 μg/adult). Zinc acetate and ZnO-NPs (without *A. mauritian*) revealed no adulticidal activity. At low doses (2 and 4 μg/adult), there was no significant (*p* = 0.09) adulticidal activity recorded. At higher doses (8, 16, and 32 μg/adult) tested materials induced significant differences (*d.f.* = 4, *p* < 0.05 and χ^2^ = 13.048). Recently, a potent larvicidal activity of ZnO-NPs synthesized from the soft coral *Ovabunda macrospiculata* was recorded against the housefly *M. domestica* with an LC_50_ of 22.595 ppm [10]. Additionally, ZnO-NPs bio-fabricated using the sea cucumber *H. impatiens* induced much higher adulticidal activity against *M. domestica* [33]. In general, *C. pipiens* larvae were more sensitive to our biosynthesized ZnO-NPs than *M. domestica* adults. Such results suggest that biosynthesized ZnO-NPs from natural origins, such as marine organisms, could exert different biological activities through the small size of NPs that makes it easily penetrate the insect cuticle.

### 2.4. Antibacterial Activity of Biosynthesized ZnO-NPs

The antibacterial efficacy of ZnO-NPs synthesized from surf redfish extracts was evaluated against two types of fish pathogenic bacteria: *Pseudomonas aeruginosa* and *Aeromonas hydrophila*. The data represented in Table 3 and Figure 3 show that ZnO-NPs were effective against the investigated bacteria, with inhibition zones ranging from 31 to 49 mm. On one side, the surf redfish extract and precursor solution (zinc acetate) did not affect the tested bacteria. On the other side, Chloramphenicol, a standard antibiotic, had a considerable effect on *A. hydrophila* with an inhibition zone of only 27 mm. According to our findings, biosynthesized ZnO-NPs exhibited promising antibacterial characteristics against fish pathogenic bacteria. Several earlier investigations have demonstrated the antibacterial action of ZnO-NPs against Gram-positive and Gram-negative bacteria, as well as antibacterial activity against spores [33,34]. Although certain assertions have been given, such as the formation of hydrogen peroxide is the key factor of antibacterial activity [35] or the binding of ZnO particles to bacterial surface owing to electrostatic forces being a mechanism, but the whole process of the antibacterial activity of ZnO-NPs are not fully known [36]. 

### 2.5. Non-Target Effects 

Water parameters were not affected by the applied concentrations of ZnO-NPs; the average water temperature was 22.41 ± 1.01 °C, dissolved oxygen was 5.32 ± 0.53 mg/L, pH was 7.05 ± 0.31, and total ammonia concentration was 0.21 ± 0.06 mg/L. 

#### 2.5.1. Gill Histopathology

Histological investigations of the gills of the non-target organism *Oreochromis niloticus* showed a normal gill histological structure in the control group. Thereafter, a series of changes were noted, including vasodilation, vacuolation, fusion of adjacent lamellae, and telangiectasia in the group exposed to the lowest concentration of ZnO-NPs (75 ppm). In addition, the fusion of adjacent lamellae, cytoplasmic vacuolation, focal necrosis, and congestion of blood vessels of the primary filaments was observed in the group exposed to 150 ppm. In the group exposed to 300 ppm, the gills were accentuated by hypotrophy and evident epithelium interstitial edema. Additionally, the fusion of adjacent lamellae with proliferation and necrosis was found in the gills of fish exposed to 600 ppm (Table 4 and Figure 4). Similar effects were observed in different histopathological deformations; hyperplasia and hypertrophy of epithelial cells, edema, lifting up the epithelial layer and congestion of blood cells, and the damage of gill epithelium due to pesticides had been noted by [37]. Hyperplasia of the gill epithelium, fusion of secondary lamellae, epithelial lifting and hypertrophy, and aneurysm were observed in the gills of the common carp exposed to chlorpyrifos [38]. These pathological changes may be a reaction to toxicant intake or could be interpreted as a general defense mechanism to prevent the entry of pollutants through the gill surface and probably due to increased capillary permeability [39]. 

#### 2.5.2. Liver Histopathology

No recognizable changes were observed in the liver of the control group. The liver of fish exposed to 75 ppm showed several histopathological changes, such as necrosis and pycnotic nuclei. The histopathological appearance of the liver following exposure to 150 ppm showed pancreatic degeneration, severe necrosis, and congestion of hepatic blood sinusoids. The fish group exposed to 300 ppm showed a congested central vein with melanomacrophages and mild focal necrosis. In addition, hepatocyte hypertrophy, moderate focal necrosis, and apoptotic cells were observed in the group exposed to 600 ppm (Table 5 and Figure 5). These histopathological alterations are in agreement with those observed in the same non-target model exposed to deltamethrin [40]. Such changes were also seen in the liver of *Cirrhinus mrigala* fish exposed to different pesticides [41].

#### 2.5.3. Hematological Parameters 

Hematological analysis revealed that our biosynthesized ZnO-NPs decreased RBCs, Hb, PCV, WBCs, and lymphocyte and monocyte counts with no effect on the heterophil count which was similar to those of the control group but increased the eosinophil and basophil counts. For serum biochemical measurements, tested ZnO-NPs decreased the serum total protein, albumin, globulin, triglyceride, and digestive enzyme levels but increased glucose, cholesterol, AST, ALT, urea, and creatinine levels as compared to those of the control. SOD and CAT levels were also elevated. Additionally, tested ZnO-NPs impacted the immune response, where lysozyme and phagocytic activity, phagocytic index, and IgM levels were decreased when different concentrations were applied (Table 6, Table 7, Table 8 and Table 9). However, ZnO-NPs were reported to induce cytotoxic effects in zebrafish embryos and larval cell lines [42], which may be attributed to the increased production of reactive oxygen (ROS) that overwhelms intracellular stress and causes cell death. Additionally, Zn^2+^ is partially released from ZnO-NPs which, in part, play a role in ZnO-NPs ecotoxicity [43]. Overall, the histopathological and hematological parameters provide an in-depth investigation into the potential side effects of ZnO-NPs against Nile tilapia fish and that the effects were dose-dependent.

## 3. Conclusions

In conclusion, an eco-friendly method was used to prepare ZnO-NPs using the sea cucumber (surf redfish) aqueous extract. The biosynthesized ZnO-NPs were characterized by UV–Vis, TEM, XRD, FT-IR, and DLS. Biosynthesized ZnO-NPs were spherical with an average size of 24.7 nm. The FTIR pattern has shown biological residues of surf redfish extract, which highlights their potential role in reducing and capping of ZnO-NPs. Larvicidal activity against *C. pipiens* and adulticidal activity against *M. domestica* were obtained after treatment with ZnO-NPs. *C. pipiens* larvae were more sensitive to our biosynthesized ZnO-NPs than *M. domestica* adults. ZnO-NPs presented a wide antibacterial activity against two fish-pathogen bacteria: *P. aeruginosa* and *A. hydrophila*. Additionally, histopathological and hematological parameter evaluations provide dose-dependent impacts of our biosynthesized ZnO-NPs against Nile tilapia fish. 

## 4. Materials and Methods

### 4.1. Sampling, Preservation and Identification of Sponge Specimen

The sea cucumber specimens were collected manually during spring 2021 from the Al-Ain Al-Sokhna coast along the Suez Gulf as described in Elbahnasawy et al. [44]. Immediately after collection, samples were washed with seawater, and then preserved in an ice box at −20 °C until processing. Identification of specimens was performed on the basis of morphological characteristics [45,46,47,48].

### 4.2. Preparation of the Aqueous Extract 

In the laboratory, the specimens were washed with tap water and then chopped into small pieces. A total of 250 gm of chopped specimens were placed in a big jar and homogenized with 500 mL of distilled water for one week; the mixture was continuously stirred with gentle shaking and stored in the dark at room temperature. Then, the mixture was filtered through a Whatman 542 filter paper [49].

### 4.3. Biosynthesis of ZnO-NPs Using Extract of Surf Redfish

The surf redfish aqueous extract and zinc acetate solution (1.5% *w/v*) (Zn (CH_3_COO)_2_2H_2_O) were mixed at a ratio of 1:1 (*v/v*), the pH was adjusted to 7.0, and the reaction was incubated (37 °C and shaken at 120 r.p.m) for 72 h in the dark [50]. The sea cucumber extract and zinc acetate solution were kept separate throughout the experiment as controls. The formation of zinc oxide nanoparticles (ZnO-NPs) was indicated by an appearance of a cloudy reaction. Purification of ZnO-NPs was carried out by washing with sterile deionized water and centrifuged at 5000 r.p.m for 15 min.

### 4.4. Characterization of Biosynthesized ZnO-NPs 

The UV–visible absorption spectra of biosynthesized ZnO-NPs were measured using a Hitachi U-2800 in the 290–710 nm range. The sample’s Fourier transform infrared (FTIR) spectra were taken in the 400–4000 cm^–1^ range using an Agilent system Cary 630 FTIR model (Agilent Technologies Inc., Santa Clara, CA, USA). The collected spectral data were compared to the reference chart to determine the functional groups present in the sample. Transmission electron microscopy (TEM) was used to examine the size and form of ZnO-NPs. For TEM analysis, a drop of the solution was placed on carbon-coated copper grids (CCG) and dried by allowing water to evaporate at room temperature. Electron micrographs were obtained using a JEOL JEM-1010 transmission electron microscope (Jeol, Tokyo, Japan) at 80 kV at The Regional Center for Mycology and Biotechnology (RCMB), Al-Azhar University, Cairo, Egypt. X-ray diffraction (XRD) was used to study the crystalline structure of the biosynthesized ZnO-NPs using the Shimadzu apparatus with nickel-filter and Cu-Ka target, Shimadzu Scientific Instruments (SSI), Kyoto, Japan. The zeta potential of the nanoparticles in the solution was determined using the Zetasizer Nano ZS, Malvern, UK. Zetasizer software (V7.12, Malvern Instruments, Malvern, UK) was used to collect and evaluate data. 

### 4.5. Insecticidal Bioassays

#### 4.5.1. Insects

*The mosquito Culex pipiens* larvae were colonized as previously described by Selim et al. [51]; they were provided *ad libitum* with fish food as a diet and maintained at laboratory conditions of 27 ± 2 °C, 70 ± 5% relative humidity (RH), with a 14–10 h photoperiod. The housefly *Musca domestica* (L.) adults were fed on a mixture of sugar and milk powder (1:1). The housefly was raised at 26 ± 1 °C with a relative humidity of 70–80% and a photoperiod of 12 h light/12 h dark cycle [52]. 

#### 4.5.2. Toxicity on *Culex pipiens* Larvae 

The larvicidal activity of synthesized ZnO-NPs using sea cucumber surf redfish (ZnO-NPs) was determined by the standard protocol of the World Health Organization [53]. The late 3rd instar larvae of *C. pipiens* were tested in five replicates of 25 larvae, with a final total of 125 larvae for each concentration. At the beginning of this experiment, a preliminary test was conducted to determine the lethal range of concentrations of ZnO-NPs. Then, larval groups were subjected to serial concentrations of (2, 4, 8, 16, and 32 ppm), alongside the negative control (distilled water) and positive control (temephos at a concentration of 1 ppm). Larval mortality was recorded 24 h post treatment. Later, lethal concentrations (LC_50_ and LC_90_) were determined and used for further non-target experiments. 

#### 4.5.3. Toxicity on *Musca Domestica* Adults 

For the adulticidal bioassay, the topical application method was used as described by Wright [54], with some modifications. Briefly, ten 3-day-old adults from both sexes were anesthetized with diethyl ether for 3 min, and then 1μL of different doses of ZnO-NPs (2, 4, 8, 16, and 32 μg/adult) were applied using a Hamilton microliter syringe 701-N (Sigma-Aldrich, Taufkirchen, Germany) on the dorsal thorax. Distilled water was used as the negative control, while cypermethrin (Dethriod 10^®^, 10% *w/v* cypermethrin, Pentacheme Co. Ltd, Bangkok, Thailand) was used as the positive control at a concentration of 1 μL/adult. After 24 h after treatment, adult mortality was recorded. Each concentration was replicated five times.

### 4.6. Antibacterial Activity of Biosynthesized ZnO-NPs Using Surf Redfish

*Pseudomonas aeruginosa* culture was kindly provided by bacteriology laboratory, Botany and Microbiology Department, Faculty of Science, Al-Azhar University. *Aeromonas hydrophilia* was kindly provided by the microbiology laboratory, Central Laboratory for Environmental Quality Monitoring and the National Water Research Center. The antibacterial activity was determined on Muller Hinton agar (MHA, Faridabad, India). On the surface of the prepared MHA, a 24-hour-old culture of bacteria (0.5 McFarland standard) was cultured. A sterile cork borer was used to cut 8 mm wells; a total of 100 µL of ZnO-NPs, aqueous extract of sea cucumber, and 1.5% (*w/v*) zinc acetate solution were applied to each well separately. Standard paper discs containing chloramphenicol were employed as the control for bacteria. The plates were left for 2 h at 4 °C, followed by incubation for 24 h at 37 °C. The experiment was repeated three times, and the inhibition zones were measured and documented after incubation [55].

### 4.7. Non-Target Effects

In this experiment, we aimed to investigate the potentially hazardous effects of biosynthesized ZnO-NPs using surf redfish against the nano-target organism *Oreochromis niloticus* following chronic exposure according to Elbahnasawy et al. [33]. Briefly, healthy adapted sex-reversed fingerling tilapia fish (average initial weight 30.21 ± 0.47 g) were exposed to concentrations of twenty-fold of obtained concentrations of larvicidal assays (75, 150, 300 and 600 ppm) in glass aquaria (40 × 25 × 30 cm, stocking density of 5 fish/aquarium) alongside the control in three replicates for each treatment according to Mount [56]. All experimental aquaria were supplied with dechlorinated tap water through a water pipeline system and supported with 24-h artificial aeration. 

#### 4.7.1. Water Quality Parameters

Several water parameters such as temperature, dissolved oxygen, pH, and total ammonia concentration were recorded throughout the experiment period. 

#### 4.7.2. Experimental Diet

Fish groups were fed a commercial diet that contained ingredients with chemical composition presented in (Table 10). Fish were fed at the rate of 3% of body weight per day at three times at 8.00, 12.00, and 15.00 h.

#### 4.7.3. Histology of Non-Targeted Organisms

At the end of the experiment, samples from each group were sacrificed with an overdose of anesthetic for conducting the histological examination of the gill and liver. Samples were fixed in 10% formalin for 24 h before being transferred into 70% ethanol for dehydration in a graded series of ethanol. Classical histological processing was carried out. Then, sections (5 μm thick) were cut and stained with hematoxylin and eosin [57].

#### 4.7.4. Hematological Assays

##### Blood Sampling and Hematological Analysis

Blood samples were collected and prepared as previously described by Elbahnasawy et al. [33]. Erythrocytes and leukocytes were counted using a hemocytometer, hemoglobin concentration was determined using the cyanomet hemoglobin method Drabkin’s solution, PCV% was estimated by micro hematocrit method, and differential leukocytic count (DLC) was calculated according to the following formula: Absolute DLC = no. of each white cell X no. of total leukocytic count/100.

##### Serum Biochemical Analysis 

Serum total protein, albumin, aspartate aminotransferase, alanine aminotransferase, triglycerides, creatinine, urea, cholesterol, and glucose were determined colorimetrically each at their corresponding wavelength, while globulin content was calculated mathematically [33].

##### Antioxidants and Other Activities

Superoxide dismutase (SOD) was colorimetrically determined at the wavelength of 560 nm, catalase at 510 nm, lipid peroxide at 534 nm, lipase at 580 nm, and amylase at 660 nm. Serum lysozyme was assayed by ELISA based on the ability of lysozyme to lyses Gram positive lysozyme sensitive bacterium with micrococcus lysodeikticus at 450 nm. Immunoglobulin (M) was measured by ELISA using a commercial kit (Bioneovan.co., Beijing, China).

### 4.8. Statistical Analysis 

Larval and adult mortality bioassays were subjected to *probit* analysis for LC_50_, LC_90,_ LD_50_, and LD_90_ at the 95% confidence limits calculation. Analysis of variance (ANOVA) and *Chi*-square values were evaluated using SPSS (ver. 25, IBM Corp., Armonk, NY, USA). The Tukey HSD post hoc test was applied for pairwise comparisons. The *p* value was measured at <0.05. 

## Figures and Tables

**Figure 1 marinedrugs-21-00437-f001:**
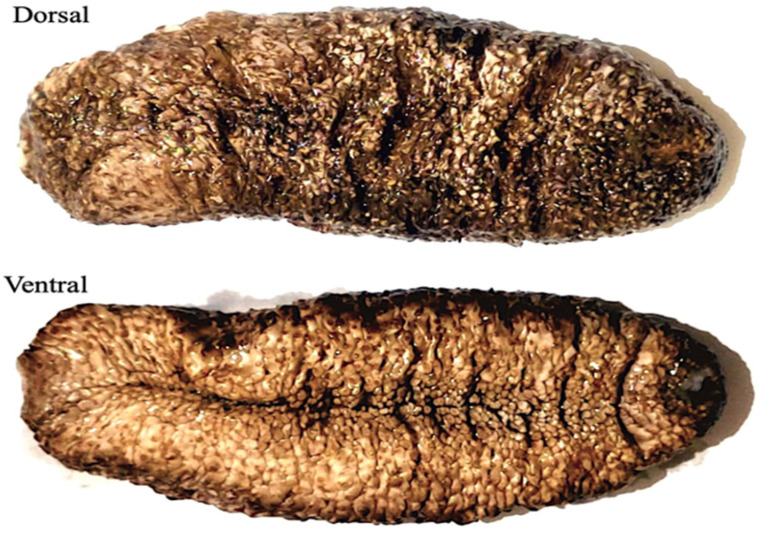
Sea cucumber; *Actinopyga mauritiana* (surf redfish).

**Figure 2 marinedrugs-21-00437-f002:**
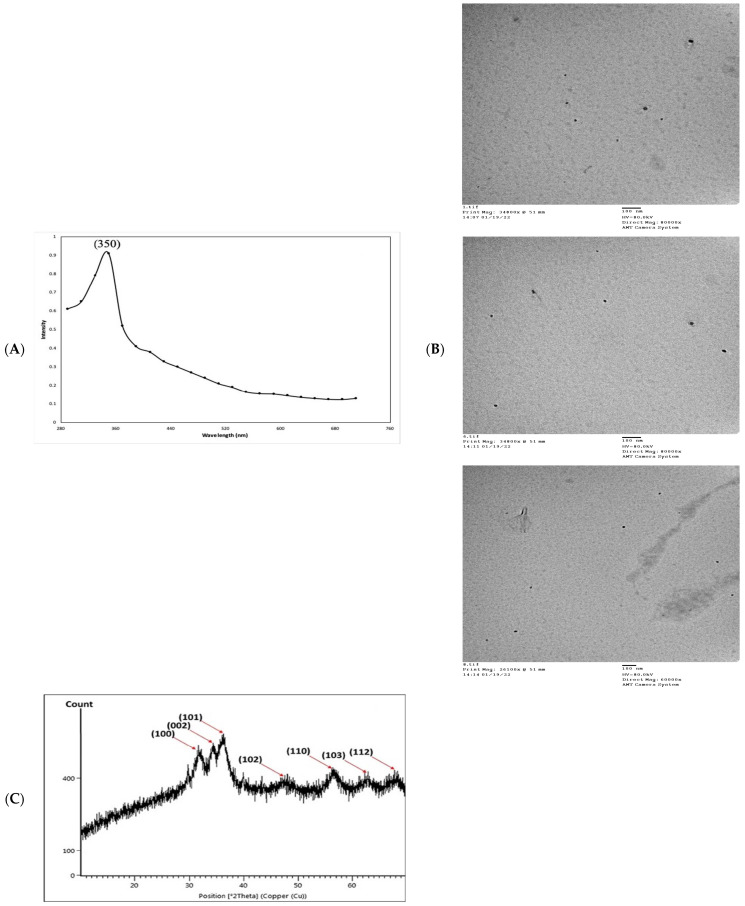
Physico-chemical characterization of biosynthesized ZnO-NPs by surf redfish extract. (**A**) UV-visible spectrum of ZnO-NPs. (**B**) TEM images of ZnO-NPs (scale bar = 100 nm), (**C**) XRD analysis of ZnO-NPs. (**D**) Zeta potential of ZnO-NPs (Mean zeta potential −0.0192 V, Standard deviation 0.0009 V, Distribution peak −0.0224 V, and Electrophoretic Mobility −1.4984 μm × cm/Vs). (**E**) FTIR analysis of ZnO-NPs.

**Figure 3 marinedrugs-21-00437-f003:**
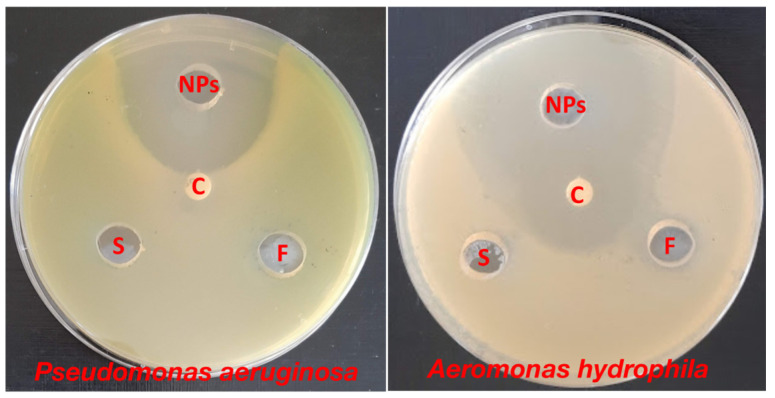
Antibacterial activity of ZnO-NPs against *Pseudomonas aeruginosa* and *Aeromonas hydrophila* using agar diffusion assay. NPs: of ZnO-NPs, C: Chloramphenicol, F: sea cucumber (surf redfish) aqueous extract, S: zinc acetate solution. Measurements were performed in triplicates for each sample.

**Figure 4 marinedrugs-21-00437-f004:**
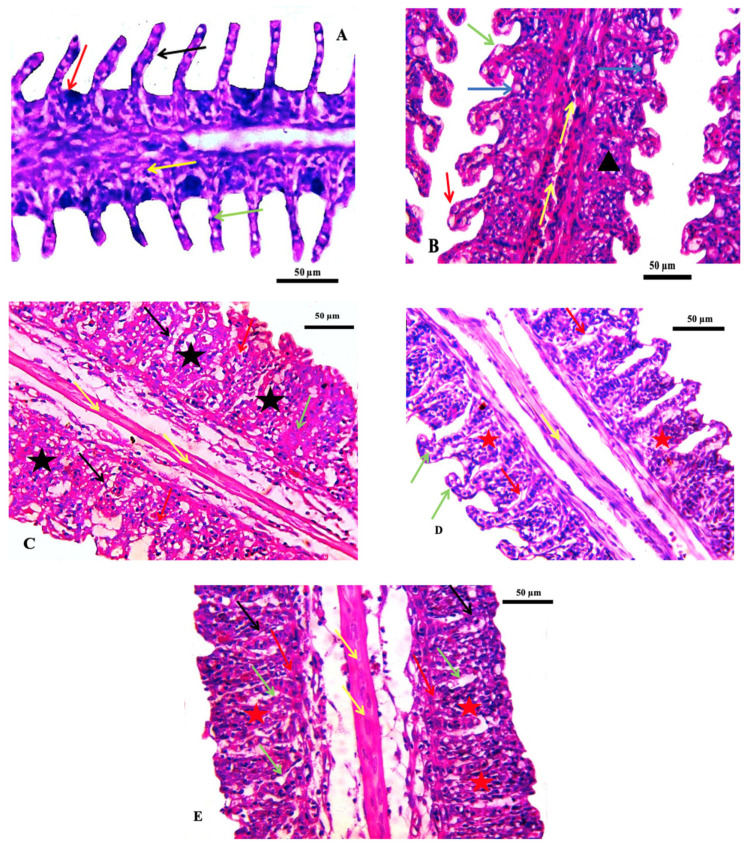
Photomicrographs of the gills of the Nile tilapia fish, *Oreochromis niloticus*. (**A**) Control gills showing secondary lamellae (black arrow), primary lamellae (red arrow) epithelial cells (green arrow) and mucous cells (yellow arrow), (**B**) Treated group by 75 ppm ZnO-NPs showing vasodilation (yellow arrow) vacuolation (blue arrow) fusion of adjacent lamellae (black arrowhead), (black arrowhead), (**C**) 150 ppm treated group indicating the proliferation of epithelium with fusion of adjacent lamellae (black star), normal central venous sinus (yellow arrow), cytoplasmic vacuolation (black arrow), focal necrosis (red arrow) and congestion of blood vessels of primary filaments filling with inflammatory cells (green arrow), (**D**) 300 ppm treated group showing hypotrophy (green arrow), mild fusion of lamellae (red star), evident epithelium interstitial edema (red arrow), and (**E**) 600 ppm treated group showing extensive fusion of adjacent lamellae with proliferation of filamentary epithelium (red star), edema (green arrow), hemorrhage (red arrow) necrosis (black arrow). Stain H&E. Scale bar = 50 μm.

**Figure 5 marinedrugs-21-00437-f005:**
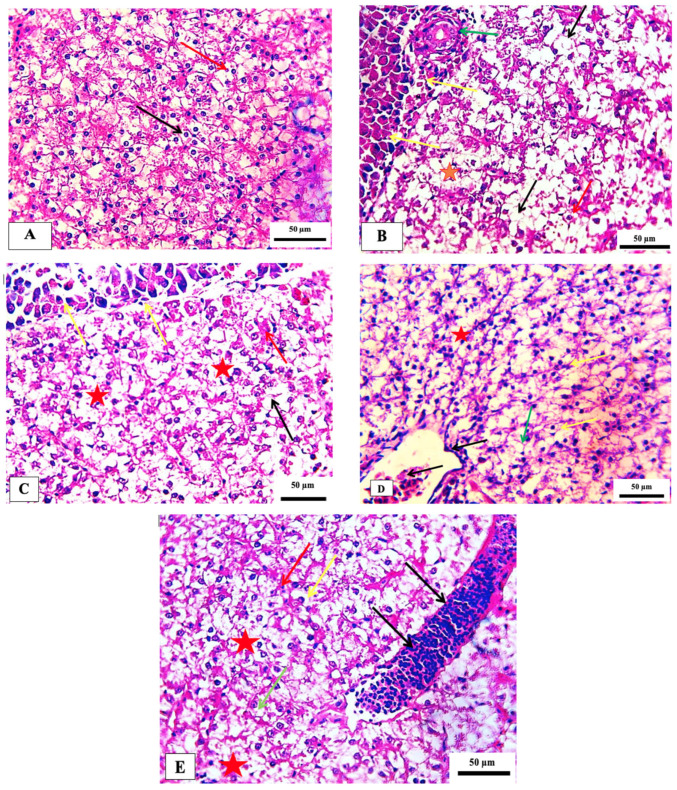
Photomicrograph of hepatopancreas of the Nile tilapia fish, *Oreochromis niloticus* showing the normal histological structure in the control group (**A**). (**B**) 75 ppm treated group showing vacuolation (black arrows), necrosis (red star), pycnotic nucleus (red arrow) with congested blood vessel (green arrow), (**C**) 150 ppm treated group demonstrating pancreatic degeneration (yellow arrow) necrosis (red star), cytoplasmic vacuolation (black arrow) and congestion of hepatic blood sinusoids (red arrow), (**D**) 300 ppm treated group showing congested central vein with melanomacrophages (black arrow), increased kupffer cell (yellow arrow), indistinct cell boundaries (green arrow) and focal necrosis (red star), and (**E**) 600 ppm treated group detecting the hepatocyte appeared hypertrophy (yellow arrow), moderate focal necrosis area (red star), apoptotic cell (green arrow) and the blood vessel noted sever dilated (black arrow). Stain H&E. scale bar = 50 μm.

**Table 1 marinedrugs-21-00437-t001:** Larvicidal activity of biosynthesized ZnO-NPs against the 3rd larval instar of the mosquito vector, *Culex pipiens*.

Concentrations (ppm)	*n*	Larval mortality % ± (SE)	LC_50_ (LCL–UCL) (ppm)	LC_90_ (LCL–UCL) (ppm)	χ^2^ (*d.f*. = 4)
Control	125	0.8 ± 0.8 ^a^	15.412 (13.679–17.580)	52.745 (42.286–70.192)	11.065 n.s.
Temephos	125	100 ± 0.0 ^f^			
2	125	3.2 ± 0.8 ^ab^			
4	125	8.8 ± 1.49 ^b^			
8	125	20.8 ± 1.49 ^c^			
16	125	47.2 ± 1.96 ^d^			
32	125	82.4 ± 1.6 ^e^			

Larval mortalities are presented as Mean ± SE of five replicates, *n* = sample size. Different letters are significantly different at (*p* < 0.05). (LC_50_) concentration that kills 50% of the population, (LC_90_) concentration that kills 90% of the population, (LCL) lower confidence limit, (UCL) upper confidence limit, (*d.f.*) degree of freedom, (χ^2^) Chi-square, n.s. = not significant (*p* > 0.05). No mortalities were recorded in the group treated with ZnO-NPs (without *A. mauritian*).

**Table 2 marinedrugs-21-00437-t002:** Adulticidal activity of biosynthesized ZnO-NPs against the housefly, *Musca domestica* adults.

Concentrations (μg/Adult)	*n*	Adult Mortality % ± (SE)	LD_50_ (LCL–UCL) (ppm)	LD_90_ (LCL–UCL) (ppm)	χ^2^ (*d.f.* = 4)
Control	50	0.0 ± 0.0 ^a^	21.132 (16.884–28.518)	84.930 (54.883–172.908)	13.048 n.s.
Cypermethrin	50	100 ± 0.0 ^f^			
2	50	2.0 ± 2.0 ^a^			
4	50	10.0 ± 1.96 ^ab^			
8	50	14.0 ± 2.45 ^b^			
16	50	32.0 ± 2.0 ^d^			
32	50	72.0 ± 4.89 ^e^			

Larval mortalities are presented as Mean ± SE of five replicates, *n* = sample size. Different letters are significantly different at (*p* < 0.05). (LC50) concentration that kills 50% of the population, (LC90) concentration that kills 90% of the population, (LCL) lower confidence limit, (UCL) upper confidence limit, (*d.f.*) degree of freedom, (χ^2^) Chi-square, n.s. = not significant (*p* > 0.05). No mortalities were recorded in the group treated with ZnO-NPs (without *A. mauritian*).

**Table 3 marinedrugs-21-00437-t003:** Antibacterial activity of ZnO-NPs synthesized from aqueous extract of surf redfish.

Bacteria	Mean ± SD of Inhibition Zone Diameter (mm) of the Tested Compounds			
	ZnO-NPs (NPs)	Surf Redfish Extract (F)	Zinc Acetate (S)	Antibiotic (C)
*Pseudomonas aeruginosa*	31 ± 3.05	0	0	0
*Aeromonas hydrophila*	49 ± 2.08	0	0	27 ± 0.57

**Table 4 marinedrugs-21-00437-t004:** Histopathological lesions recorded on the gills of Nile tilapia fish (*Oreochromis niloticus*) exposed to different concentrations of biosynthesized ZnONPs.

Type of Gill Change	Control	Treatments (ppm/mL)			
		75	150	300	600
Lamellar fusion	−	+	+	+	++
Epithelium lifting	−	−	+	+	+
Vasodilatation	−	+	−	−	−
Aneurisms	−	+	−	−	−
Edema	−	−	+	+	++
Necrosis	−	−	+	−	+
Cytoplasmic vacuolation	−	+	+	−	−

(−) not alterations found, (+) visible alterations, (++) medium alterations.

**Table 5 marinedrugs-21-00437-t005:** Histopathological lesions recorded on the liver of Nile tilapia fish (*Oreochromis niloticus*) exposed to different concentrations of biosynthesized ZnO-NPs.

Type of Liver Change	Control	Treatments (ppm/mL)			
		75	150	300	600
Cytoplasmic vacuolation	−	+	+	+	+
Pancreatic degeneration	−	+	+	−	−
Pyknotic nuclei	−	+	+	+	+
Necrosis	−	+	+	+	++

(−) not alterations found, (+) visible alterations, (++) medium alterations.

**Table 6 marinedrugs-21-00437-t006:** Hematological parameters of the Nile tilapia fish, *Oreochromis niloticus* exposed to different concentrations of biosynthesized ZnO-NPs.

Parameters	Treatments (ppm)				
	0	75	150	300	600
RBCS (×10/mm³)	4.12	3.18	3.31	2.79	2.82
HBg/100ml	12.6	9.6	10	8.45	8.59
PCV%	40	31	33	27	28
MCV(μm³/cell)	97.09	97.48	99.7	96.77	99.29
MCH (pg/cell)	30.58	30.19	30.21	30.29	30.46
MCHC	31.5	30.97	30.3	31.3	30.68
WBCs (×10³/mm³)	10.76	10.19	9.98	10.32	10.2
Heterophil (×10³/mm³)	2.15	2.75	2.2	2.48	1.94
Lymphocyte (×10³/mm³)	7.53	6.42	6.59	6.91	7.34
Monocyte (×10³/mm³)	0.86	0.71	0.7	0.72	0.71
Basophil (×10³/mm³)	0.11	0.1	0.2	0.10	0.1
Esinophil (×10³/mm³)	0.11	0.2	0.3	0.10	0.1

**Table 7 marinedrugs-21-00437-t007:** Immune responses of the Nile tilapia fish, *Oreochromis niloticus* exposed to different concentrations of biosynthesized ZnO-NPs.

Parameters	Treatments (ppm)				
	0	75	150	300	600
Lysozyme (µg/mL)	6.29	6.05	4.85	3.99	5.16
Phagocytic index (%)	1.3	1.2	1.22	1.09	0.99
Phagocytic activity (%)	12.24	8.15	9.23	10.06	11.64
IgM (µg/mL)	5.26	3.99	2.89	4.86	5.01

**Table 8 marinedrugs-21-00437-t008:** Serum biochemical measurements of the Nile tilapia fish, *Oreochromis niloticus* exposed to different concentrations of biosynthesized ZnO-NPs.

Parameters	Treatments (ppm)				
	0	75	150	300	600
Glucose (mg/dL)	10.25	14.86	15.67	14.66	15.95
Cholesterol (mg/dL)	69.86	79.86	85.36	90.01	79.64
Triglyceride (mg/dL)	103.15	82.65	94.33	99.87	100
Total protein (g/dL)	4.19	3.69	3.51	3.78	3.71
Albumin (g/dL)	1.66	1.41	1.5	1.62	1.53
Globulin (g/dL)	2.53	2.28	2.01	2.16	2.18
AST (U/L)	30.11	40.19	38.25	41.26	29.86
ALT (U/L)	30.16	40.12	27.98	30.18	31.45
Urea (mg/dL)	2.19	2.64	2.41	2.38	2.27
Creatinine (mg/dL)	0.4	0.52	0.43	0.44	0.49
Lipase (U/L)	29.86	29.15	28.75	29.98	27.02
Amylase (U/L)	61.01	44.98	51.32	39.86	40.58

**Table 9 marinedrugs-21-00437-t009:** Serum antioxidant biomarkers of the Nile tilapia fish, *Oreochromis niloticus* exposed to different concentrations of biosynthesized ZnO-NPs.

Parameters	Treatments (ppm)				
	0	75	150	300	600
MDA nmol/g	16.54	26.53	17.64	29.32	23.02
CAT U/gm	18.17	13.85	17.64	15.02	16.88
SOD U/gm	23.58	16.94	15.89	18.98	20.15

**Table 10 marinedrugs-21-00437-t010:** Diet formulation and proximate analysis (g/kg).

**Diet ingredients%**	
Fish meal	18.0
soybean meal	29.0
yellow corn	20.0
wheat bran	15.0
alfalfa hay	12.0
sunflower oil	3.0
minerals mixture	1.0
vitamin mixture	1.0
carboxymethyl cellulose	1.0
Total	100
**Chemical composition (g/kg)**	
crude protein	30.11
ether extract	12.35
Ash	14.34
NFE ^1^	43.20
GE ^2^	4600 Kcal/kg

^1^ NFE, nitrogen free extract = 100 − (CP+ CF+ EE+ Ash %). ^2^ GE, gross energy calculated using the 5.65, 9.45, and 4 for CP, EE and NFE, respectively.

## Data Availability

We will provide all data generated in this study upon request.

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
