# Peer review of "Surf Redfish-Based ZnO-NPs and Their Biological Activity with Reference to Their Non-Target Toxicity"

_marinedrugs, 2023, doi:10.3390/md21080437_

Round 1

Reviewer 1 Report

The study aimed to evaluate the larvicidal, adulticidal, and antibacterial activities and the toxicity to the non-target organism of sea cucumber-fabricated zinc oxide nanoparticles (ZnO-NPs). Besides providing the physic-chemical characterization, the authors have evaluated its potential as anti-Diptera. Antibacterial and the potential side-effects of these ZnO-NPs on Nile tilapia (Oreochromis niloticus). The study is complete, but a few minor revisions should be addressed before publication.

Figure 1. The pictures are not good quality. Could the authors give a better figure? I will even suggest using a picture of the animal in the water.

In several sections of the manuscript (2.3 and 2.4), I miss some explicit sentences which assert the potential use of these ZnO-NPs. Please, add them.

The Conclusions section should be placed after the Results and Discussion section.

Author Response

Response to reviewer (1) comments

First of all, we really appreciate the reviewer’s valuable comments that shaped our manuscript better and added true value to the presented work, as well as the handling editor for his kind consideration of this article for possible publication.

Comments.

The study aimed to evaluate the larvicidal, adulticidal, and antibacterial activities and the toxicity to the non-target organism of sea cucumber-fabricated zinc oxide nanoparticles (ZnO-NPs). Besides providing the physic-chemical characterization, the authors have evaluated its potential as anti-Diptera. Antibacterial and the potential side-effects of these ZnO-NPs on Nile tilapia (Oreochromis niloticus). The study is complete, but a few minor revisions should be addressed before publication.

Minor comments

  • Figure 1. The pictures are not good quality. Could the authors give a better figure? I will even suggest using a picture of the animal in the water.

Response

Thanks a lot for your valuable comment. We have replaced it with another one with better resolution.

  • In several sections of the manuscript (2.3 and 2.4), I miss some explicit sentences which assert the potential use of these ZnO-NPs. Please, add them.

Response

Thanks for your valuable comment. We have added several sentences in this context in the revised manuscript.

  • The Conclusions section should be placed after the Results and Discussion section.

Response

Thanks for your comment.  We have moved it, thanks. 

********

Reviewer 2 Report

The synthesis of nanomaterials by biological methodologies has always been interesting in order to use the
least amount of polluting agents. However, I have some observations that should be taken into consideration.
    1.  Reference citations must be in order. On line 130 the authors cite reference 50 after reference 22.
Correctly order the references that are cited in the text.
2.  All figures must be improved, the texts on the X axis and the Y axis must be visible to the reader.
Use a homogeneous format for all graphs.
3.  The TEM image of the nanoparticles is not clear, I would like to verify a higher resolution image.
The scale legends are not clearly visible, so the reader can verify the diameter from the image.
I suggest placing another TEM image.
4.  The antimicrobial evaluation should be repeated, since it is not conclusive. Although it is to
demonstrate that Zinc Oxide (ZnO) nanoparticles and their antimicrobial and larvicidal effect
were synthesized. Authors should carry out the assay in triplicate, eg with the positive control
and negative control. In Figure 3 they could only show 3 images of Petri dishes, a Petri dish where
they put the positive control (antibiotic) in triplicate with a negative control sample (distilled water).
A Petri dish where 3 ZnO-NP samples and the negative control are placed. A Petri dish with the negative controls.
5.  It is not necessary to evaluate the sea cucumber extract, since it was used only as a reducing agent
(This would be better seen in the UV-VIS graph), as well as the Zinc acetate solution, since this reagent was used
as a precursor agent (This would look better on the UV-VIS plot.
This means that in Figure 2, the UV-Vis spectrum
of the extract, the UV-Vis extract of the precursor agent and the spectrum of the ZnO nanoparticles could be observed.

Author Response

 Response to reviewer (2) comments

Deep thanks to the reviewer’s valuable comments that shaped our manuscript better and added true value to the presented work.

Comments.

The synthesis of nanomaterials by biological methodologies has always been interesting in order to use the least amount of polluting agents. However, I have some observations that should be taken into consideration.     

  1. Reference citations must be in order. On line 130 the authors cite reference 50 after reference 22 Correctly order the references that are cited in the text.

Response

We have corrected it to (5), thanks.

  1. All figures must be improved, the texts on the X axis and the Y axis must be visible to the reader. Use a homogeneous format for all graphs.

Response

Thanks a lot for your valuable comment. We have replaced them with others but the quality is still low (Sorry for that).

  • The TEM image of the nanoparticles is not clear, I would like to verify a higher resolution image. The scale legends are not clearly visible, so the reader can verify the diameter from the image. I suggest placing another TEM image.

Response

Electron micrographs were obtained by JEOL JEM-1010 transmission electron microscope which is not HR TEM. We replaced the TEM image with three images at different fields. The scale legends were mentioned in the caption, thanks.

  • The antimicrobial evaluation should be repeated, since it is not conclusive. Although it is to demonstrate that Zinc Oxide (ZnO) nanoparticles and their antimicrobial and larvicidal effect were synthesized. Authors should carry out the assay in triplicate, eg with the positive control and negative control. In Figure 3 they could only show 3 images of Petri dishes, a Petri dish where they put the positive control (antibiotic) in triplicate with a negative control sample (distilled water).  A Petri dish where 3 ZnO-NP samples and the negative control are placed. A Petri dish with the negative controls.

Response

We appreciate the reviewer for this careful note. We measured the antibacterial activity of ZnO-NPs and controls against Pseudomonas aeruginosa and Aeromonas hydrophila.  The assays were caried out in triplicate. Fig 3 shows Petri dishes containing ZnO-NPs, two negative controls (sea cucumber aqueous extract and zinc acetate solution), and one positive control (antibiotics). Although that were done three times independent. 

  • It is not necessary to evaluate the sea cucumber extract, since it was used only as a reducing agent (This would be better seen in the UV-VIS graph), as well as the Zinc acetate solution, since this reagent was used as a precursor agent (This would look better on the UV-VIS plot.This means that in Figure 2, the UV-Vis spectrum of the extract, the UV-Vis extract of the precursor agent and the spectrum of the ZnO nanoparticles could be observed. 

Response

Thanks a lot for your valuable comment. We have shown that in multiple publications, please see the following two recent articles. https://doi.org/10.3390/separations10030173; https://doi.org/10.3390/jfb13040242 

*********
